# Infancy Dietary Patterns, Development, and Health: An Extensive Narrative Review

**DOI:** 10.3390/children9071072

**Published:** 2022-07-18

**Authors:** Alexandra Martín-Rodríguez, Álvaro Bustamante-Sánchez, Ismael Martínez-Guardado, Eduardo Navarro-Jiménez, Erika Plata-SanJuan, José Francisco Tornero-Aguilera, Vicente Javier Clemente-Suárez

**Affiliations:** 1Faculty of Sports Sciences, Universidad Europea de Madrid, 28670 Madrid, Spain; sandra.martin.rodriguez8@gmail.com (A.M.-R.); vctxente@yahoo.es (V.J.C.-S.); 2Facultad de Ciencias de la Vida y la Naturaleza, Universidad de Nebrija, 28240 Madrid, Spain; imartinezgu@nebrija.es; 3Facultad de Ciencias de la Salud, Universidad Simón Bolivar, Barranquilla 080002, Colombia; enavarro27@unisimonbolivar.edu.co; 4Independent Researcher, Barranquilla 080002, Colombia; erikaplata.nutricion@gmail.com; 5Grupo de Investigación en Cultura, Educación y Sociedad, Universidad de la Costa, Barranquilla 080002, Colombia

**Keywords:** nutrition, children, infancy, dietary, patterns, progenitor

## Abstract

Correct dietary patterns are important for a child’s health from birth to adulthood. Understanding a child’s health as a state of entire physical, mental, and social well-being is essential. However, reaching adulthood in a complete health proper state is determined by feeding and dietary habits during preconception, pregnancy, or children infancy. Different factors, such as the mother’s lifestyle, culture, or socioeconomic status, are crucial during all these phases. In this review, we aimed to assess the long-term associations between infancy dietary patterns and health and their influence on development and growth. To reach this objective, a consensus critical review was carried out using primary sources such as scientific articles, and secondary bibliographic indexes, databases, and web pages. PubMed, SciELO, and Google Scholar were the tools used to complete this research. We found that high-income countries promote high-calorie foods and, consequently, obesity problems among children are rising. However, undernutrition is a global health issue concerning children in low- and middle-income countries; thus, parental socioeconomic status in early life is essential to children’s health and development, showing that biological, social, and environmental influences are increased risk factors for chronic diseases. This narrative review is aimed to collect evidence for early nutritional intervention and future disease prevention.

## 1. Background

Better health during infancy and childhood, a stronger immune system, safer pregnancy and childbirth, lower risk of metabolic diseases, and having good nutrition are all related to improving your diet. Thus, eating a variety of foods is essential for a healthy diet. Nevertheless, it has been reported in 2020 that 149 million children were estimated to be stunted (too short for age), 45 million were estimated to be wasted (too thin for height), and 38.9 million were overweight or obese [1,2]. Evidence suggests that these nutritional deficiencies were developed throughout infancy, and cognition was affected even during adulthood [3]. This malnutrition is related to the family system that surrounds a child’s domestic life, which has an active role in establishing and promoting behaviors [4,5]. It has been demonstrated that parental food habits and feeding strategies make a difference in children’s eating behaviors and food choices [6]. Additionally, different factors associated with the progenitor’s culture have been investigated regarding nutrition in infancy [7,8]. In order to make this topic more concrete, this review presents the processes that could influence the child’s life from preconception through the first years of life. Although it is demonstrated that changes in nutrient quantity and quality during these periods may permanently influence health [9], more knowledge is needed to implement nutrition interventions.

There is considerable evidence accumulated, especially during the last decade, demonstrating that early nutrition and lifestyle have long-term effects on later health and disease [10]. Particularly during preconception and pregnancy, it is acknowledged that a focus on parents’ health offers an important opportunity for improving the health of future generations [11]. The International Federation of Gynecology and Obstetrics recommendations report that good health and nutrition before conception are central to a mother’s ability to meet the nutrient demands of pregnancy and breastfeeding, and are vital to the healthy development of her embryo, fetus, infant, and child [12]. On the basis of this evidence, there should be a concerted attempt to develop interventions to help women achieve a healthy weight before pregnancy [13]. After birth, breastfeeding is associated with numerous benefits and is universally recommended as the preferred method of infant feeding [14]. For all further growth, both under- and overfeeding should be avoided, and energy and nutrient intakes should be adapted to achieve a weight gain similar to the normal weight gain defined by generally accepted growth standards [15]. When nutrition interventions were implemented, different results were achieved [16]. Therefore, in order to analyze parents’ nutritional habits and infant feeding and its influence on their growth and health, we conducted the present research.

### Search Methods and Strategies for Research Identification

The protocol used consisted of a literature search, using primary sources, such as scientific articles, and secondary, such as bibliographic indexes, databases, and web pages. We used PubMed, Embase, SciELO, Science Direct Scopus, and Web of Science, employing MeSH compliant keywords, including infancy dietary patterns, nutrition during infancy, nutrition and cognition, dietary patterns and children’s development, pregnancy nutrition, nutritional socioeconomic status, progenitor’s dietary patterns, and feeding patterns. We used articles and information published from 2014 to 2022, although previous studies from the 2000s and earlier were included to explain some information in several points of the review. The following exclusion criteria were used: (i). studies with inappropriate or not relevant topics, being not pertinent to the main focus of the review, (ii). Ph.D. dissertations, conference proceedings, and unpublished studies. We included all the articles that met the scientific methodological standards and had implications with any of the subsections in which this article is distributed: nutrition, development, and infant´s health. The information treatment was performed by the seven authors of the review, and each sub-topic was decided by consensus. Finally, articles were discussed by the authors to write the present review. The final 228 papers were read to be considered relevant to the search criteria and appropriate for assessing our research objective.

## 2. Progenitors Culture and Dietary Patterns in Infancy

Culture is an integral component of food habits, affecting what, when, and how we eat [17]. These habits are achieved mostly through transference in parental guidance [18]. Thus, eating behaviors have been investigated over the years to develop healthy habits and command children’s nutrition for preemptive intentions. However, it seems to be a multifactorial context where progenitors’ culture and their domestic life play a role in setting up and promoting practices that will persist throughout the lifespan, not only in infancy [5]. Although preferences can change, children are also under the influence of biological, social, and environmental factors where their eating-related attitudes are built [19]. These attitudes could have crucial implications, such as undernutrition or, contrarily, increased obesity risk [20]. Even though progenitors’ practices should be appropriated to prevent unhealthy eating patterns, unfortunately, it is not always the case.

Regarding food selection, the relationship between nutrition and culture has been demonstrated previously [21]. This relationship is based on people’s beliefs and preferences. It is known that people can do an inadequate food selection and, therefore, have bad eating habits despite the fact all kinds of foods are available in their countries. For this reason, it is not surprising to find differences in food selection between countries, as well as between different locations in the same country. In fact, in the last decades, a phenomenon of nutrition transition has been developed, and it has been defined as the shift from the traditional to a ‘‘Westernized’’ diet. Evidence suggests that food choices are affected by demographic transition [22]. However, regardless of location, a child’s early experience influences food likes later in life [23]. After birth, in some cultures, children are deliberately exposed to strong flavors. For example, Mexican culture uses chili peppers to increase strength in food. Learning to like initially strong tastes may be part of the socialization process [5]. Moreover, in Western cultures, when tofu, or plain, salted, or sweetened foods are given to preschool children repeatedly, they came to prefer the version that had become familiar to them [24], which suggests that repeated exposure to food increases their acquaintance, and it is one of the substantive determinants of their acceptance.

Findings have also revealed that food choices and even new food practices are adopted by migrants when they start a new life in a new country as a part of their inclusion process [25,26]. Though traditional practices are maintained, they include new foods in their diet to generate a connection with the new location [24,27,28,29]. Therefore, due to mobility, urbanization, and every attempt to belong to the new culture, low-income immigrant families spend a lot of time away from home and are unable to establish healthy eating habits; thus, malnutrition and obesity have become major risks [29]. The latter phenomenon is also due, in part, to dietary acculturation, signaled by the adoption of the eating and consumption patterns of the host country. It is now widely recognized that the Western diet is deficient in nutrient-rich foods such as fruits, vegetables, and whole grains, and superfluous in energy sources of solid fats, added sugars, and alcoholic beverages [30].

Food selection and dietary habits are often conditioned by food allergies. Whether a child is more or less likely to suffer from a food allergy can be determined by family culture. It has been demonstrated that children of immigrants (second-generation immigrants) have been suggested to be at high risk of food sensitization. In addition, environmental changes in the microbiome and/or diet due to migration to industrialized and/or Western countries may contribute to atopy presentation [31]. Even migrants and natives living in the same geographical location have differing allergy prevalence that forces them to adjust their dietary habits. Increased populations living in cities, migration, and economic growth have led to an increase in the incidence of food allergies [32]. Studies show that the environment and ethnicity play a part in this [33].

Further research is needed to understand how progenitor culture affects infants´ dietary habits. Considering that societies are currently a mixture of traditional cultures and Western culture, as well as the results presented, we would suggest, as future research lines in this area, the study of the effect of this culture exchange on infant feeding. Knowing these mechanisms, pediatricians could develop effective nutrition programs and create healthy food practices among the child population.

## 3. Socioeconomic Status of Progenitors and Dietary Patterns in Infancy

An unhealthy diet is a well-known risk factor that can lead to chronic non-communicable diseases such as obesity and hypertension. It should be noted that it is crucial not only to focus on the poor nutrition of children, but also on the poor nutrition of parents. The majority do little to change their lifestyle to prepare for pregnancy, and socioeconomic status plays a part in it. Pre-pregnancy Body Mass Index (BMI) (body weight (kg)/height^2^ (m)) and preconception supplementation are strongly related to health outcomes. This fact is important for fetal development; however, implementing healthy habits is not accessible to all [34]. Furthermore, both parents’ health is relevant, and some studies demonstrated that dietary zinc deficiency impairs reproduction in males and females [35]. It is estimated that one-third of the world population has a zinc shortage [36]. Particular attention should be also paid to the intake and status of some other micronutrients, especially folate, in women of reproductive age. Studies show strong links between health before pregnancy and maternal and child health outcomes, with consequences that can extend across generations [4]. Concretely, dietary supplementation with iron, vitamin D, vitamin B12, iodine, and others may be indicated in women at risk of poor supply and insufficiency of these micronutrients. Globally, preconception use of folic acid is estimated to be under 50% [37], with particular concern that young women from lower socio-economic backgrounds are the least likely to follow the recommendations [38].

As mentioned above, the economic status (SES) plays an important role in this process, and further research is needed in countries with resource-limited countries. Research in high-income countries partly attributes disparities in obesity and other health conditions to differences in dietary quality [39,40]. In most high-income countries, high-calorie foods are cheaper, whereas healthier foods tend to be more expensive [41,42], so this will be a crucial factor in determining diet quality [43,44]. Due to this, childhood obesity has increased, and it is now recognized as a global public health problem [45]. Several authors have confirmed a high incidence of overweight and obesity among children aged 5–18 years in various parts of the world [46,47,48]. Findings have also reported a prevalence of overweight among children ranging from 11.8% to 16.33%, whereas the prevalence of obesity was moderately lower, ranging from 4.9% to 10.69% [45,46,47,48].

In contrast, undernutrition is a global health issue concerning children in low- and middle-income countries (LMICs). Approximately one in five children worldwide suffers from childhood malnutrition and its complications, including increased susceptibility to inflammation and infectious diseases [49]. Even so, Coetzee et.al found in a 7-year longitudinal study conducted in 181 South African school children (63 Caucasian and 118 black and mixed-race) that BMI increased over the period studied (2010–2016). Children in higher SES groups were more likely to be overweight and obese (compared to lower socio-economic status groups (overweight: 1.09–2.17% and obese: 2.17–4.35%)). The results further indicate that although there was a decrease in obesity in high-SES children, its prevalence was still higher than in low-SES children [50].

In conclusion, studies show that low-income groups are more likely than high-income groups to have a higher unhealthy diet pattern and lower healthy diet pattern scores [51], which may be related to the fact that unhealthy foods generally cost less and, therefore, may be the only affordable option for low-income groups. In addition, the positive relationship between healthy eating and income may be explained by the fact that healthier diets have been reported to cost more; therefore, those in high-income groups are more likely to indulge in this pattern of eating, including children [44,52].

## 4. Mother’s Diet during Pregnancy and Baby’s Health

To point out the factors that determine a correct dietary pattern during childhood, it is necessary to address the gestation process. In this line, it seems that to reduce health problems and the risk of suffering from chronic diseases during the early stages of life, it is very important to control the nutritional and metabolic environment during pregnancy [53].

The development of the child is affected by the diet and exercise habits of the pregnant woman [54,55]. Having a healthy lifestyle may prevent complications during pregnancy, as well as prevent the newborn from developing non-communicable diseases [56]. Due to this evidence, parents’ habits will possibly alter the whole life of their children [57]. Overweight and obesity have been linked to most diseases [52]. Fetal macrosomia and low birth weight have been associated with gestational diabetes in studies with overweight and obese mothers [58,59].

Thus, findings suggest that this maternal overweight or obesity may also lead to a lower life expectancy for the child, as well as miscarriages and premature births [60,61,62,63]. It has been shown that a high-calorie diet is detrimental to both the mother and the child’s health [64,65]. Therefore, during the pre-gestational period, mothers should try to maintain an adequate weight and healthy eating habits, since during the gestational period, their BMI will largely determine many of the consequences later on [66,67]. It has been shown that even a 10% weight loss in obese women can be a crucial factor for the development of the child. Regarding evidence and general recommendations, a balanced diet is required [67]. The increased consumption of vegetables and fruits, as well as wholegrain products, legumes, and fish showed a lower risk of gestational diabetes is needed to prevent diseases. Additionally, it should be noted that nutrient-dense foods are favorable during gestation because more energy is absorbed. In pregnant women, three servings of vegetables and two servings of fruit should be implemented every day [68]. Foods such as cereal products, especially whole grains, and potatoes are rich in vitamins, minerals, and fiber. Protein, calcium, and iodine may be provided by milk and dairy products [69]. Finally, fish is an important source of vitamin B 12, zinc, and iron. All of them are essential components of a balanced diet. Measures to achieve this should be indicated by medical personnel following scientific evidence [70].

A balanced diet, regular exercise, and a healthy lifestyle are very important before and during pregnancy. The time before conception and the first 1000 days of the child’s life provide the opportunity to lay the foundation for the health of the child and the mother-to-be. The revised recommendations presented here provide practical and up-to-date knowledge-based recommendations for pregnancy and also for women/couples wishing to have a child [57].

## 5. Nutrients Intake in Infancy and Child Development

Early child development is a crucial period for ensuring an individual’s physical and mental health, with nutrition being among the most important risk factors for early child development [71]. The latest estimates suggest that around 250 million children under five in low- and middle-income countries were at risk of not achieving their full development potential due to stunting and extreme poverty [72]. Contrary, in industrialized countries, child obesity is an important public health challenge, with the most recent data indicating that approximately 18.5% of the population from 2–19 years of age are obese in the United States alone, and current trends in the United Kingdom suggest that by 2050, 25% of all children under 20 years of age will be classified as obese [73,74].

Therefore, an optimal balance between micro and macronutrients is necessary for optimal development and to be able to avoid typical Western diseases in the future; thus, nutrition is essential to children’s future. Industrialized countries are more than sufficient to meet physiological requirements among children [75], but it is also easy to exceed it, and a high protein intake in early childhood has been linked to a higher risk of obesity [76]. One study showed high protein intake in early childhood to be associated with a higher fat mass index (FMI), but not with a higher fat-free mass index at school age. This association is stronger from animals than from vegetable protein [77]. An official upper limit of protein intake is yet to be established among children.

Regarding micronutrients (vitamins and minerals), research on the relationship between micronutrient status during early childhood and obesity in later life is urgently needed. Growth (weight and length) during the first 2 years of life shows the nutritional status of infants and young children is the consequence of breastfeeding and complementary feeding [78]. Particularly, a review carried out by Singhal et al. revealed that rapid weight gain in infancy is positively associated with obesity in later life [79]. A slower rate of weight gain and possibly a decreased risk of overweight in childhood and adolescence compared to formula feeding have been related to exclusive breastfeeding [78]. Breastfeeding is associated with ~a 20% reduction in the odds of being overweight, whereas a lack of breastfeeding, low birth weight, and rapid weight gain were associated with obesity [79].

## 6. Dietary Patterns in Infancy and Cognitive Function

Good cognitive development and brain function during the prenatal period and early years is influenced by nutrition [80]. Findings have suggested that children with a better nutritional status could have improved their cognitive and neuropsychological function [81]. The quality of nutrition during pregnancy and breastfeeding is reported to be relevant for better test scores of cognitive functions [82]. Furthermore, studies have looked at the relationship between early nutrition status and growth in infancy and childhood. Knowing these two aspects and the complex interactions between environmental stimuli and nutritional patterns, this issue should be further explored in future research.

Children´s growth is determined by their early nutrition status. Evidence has established that early nutrition can have a long-term effect on growth, metabolic outcome, and long-term health [83]. This status depends on many factors, including the mother’s diet [84], her socioeconomic status [84], and if her children are breastfed or not. Thus, a balanced maternal diet during pregnancy is essential. The formation of the neural tube depends on these nutrients, and a deficiency can adversely affect brain development, resulting in neural tube defects, spina bifida, and encephalocele [85]. Moreover, iron is necessary for neurogenesis and dopamine production. A deficiency in these nutrients may cause significant cognitive impairment in the offspring [86,87,88]. Concerning breastfeeding, neurological benefits of breastfeeding in infancy were demonstrated previously [85,89]. When the supply of long-chain polyunsaturated fatty acids through the placenta is interrupted, this supplementation depends on the mother´s diet [90]. The evolution of the mother’s body has created mechanisms that adjust the amount of fat in milk to the needs of the child [91]. Knowing that breast milk contains long-chain polyunsaturated fatty acids which form the major 34 structures of 35 neuronal membranes, it is important to highlight that it could play a critical role in human nervous system functioning [92,93]. Firstly, there is considerable evidence linking breastfeeding with better performance in intelligence tests [94]. Studies show that three months of breastfeeding improves intelligence quotients 2.1 times over others [82]. Additionally, improvements in motor skills, as well as language development, have also been found [95]. Even if breastfeeding is extended to 6 months, there seems to be a lower risk of developing attention deficit, hyperactivity, or autism spectrum syndrome [96]. However, other studies have found no difference between the promotion of exclusive breastfeeding and children aged 5–8 years on several measures of children’s cognitive development [97,98]. Although diets have been extensively studied concerning children’s cognitive development, there is less research showing interest in the transition from liquid foods to solids. Therefore, further research is needed to understand how the influence of context and socioeconomic status may influence cognitive functions. However, evidence suggests that the promotion and support of breastfeeding and other healthy feeding practices are especially important for children of low socioeconomic status, who are at increased risk of obesity [99] and cognitive impairment [100].

Regarding the nutritional status, the degree of nutrient deficiency may be modified or even increased during the mother’s and child’s lifespans. Findings suggest that certain types of deficiency, such as iron deficiency, are related to impaired brain development [3]. All the influences surrounding both mother and child have attributed a relevant role in this process. The rapid growth of the brain during the gestational period makes it very vulnerable to an inadequate diet. Comparing both types of countries, developing and industrialized, evidence in Kenya and Mexico showed that where the mother’s energy intake declines gradually throughout pregnancy, not only do mothers gain only half as much as European or North American women, but also, they even lose weight and fat in the last month of pregnancy [101]. In both, iron scantiness occurs commonly, which is known to adversely influence cognition [80,102]. Although maternal nutrient deficiencies could be modified with micronutrient supplementation starting in pregnancy, increasing birth weight, there is a burgeoning importance of maternal health before conception and the key risk factors for unfavorable birth outcomes, such as those related to cognitive development. There is a need to develop programs and policies to enhance nutritional status across the life course and especially during reproductive ages to promote healthy patterns in infancy. Moreover, the importance of health before and after pregnancy and possible actions to take may contribute to this [4].

## 7. Dietary Patterns in Infancy and Body Composition

Currently, to dig into the effects of diet on health, the scientific community has focused on observing dietary patterns in general, rather than examining particular macro- or micronutrients in isolation [103]. Though the gestation process plays an important role in determining a child´s BMI, there are other factors, such as diet composition, to take into account. It has been established that the rapid growth and increase of adiposity are one of the main phenomena that occur during childhood [104]. Thus, several studies have reported that adopting unhealthy dietary patterns, characterized by energy-dense, low-fiber, and high-fat foods, during early childhood presents a strong relationship to developing obesity in later childhood [105]. Therefore, other than increasing the risk of suffering obesity in adulthood, this can cause other types of coronary problems, diabetes, and premature death [106].

According to Leermakers et al. (2018) [107], childhood dietary patterns should be explained under practical approaches, such as variation in dietary intake and specific biomarkers related to health at this stage, which could predict variations in childhood body composition (i.e., fat mass and fat-free mass index) [108]. In line with this, the dietary pattern used with growing children will determine the development of adequate body composition values, decreasing the level of adiposity by reducing fat mass and/or increasing fat-free mass [109]. Thus, Robinson et al. reported higher values of lean body mass in 4-year-old children who had been fed a diet based on fruits, vegetables, meat, and fish, and other foods prepared at homes, such as rice and pasta, during weaning. As has been previously established, a higher lean mass has been associated with improved cardiovascular and metabolic health [110,111]. Moreover, the use of these types of healthy dietary patterns has reported decreases in body weight, BMI, or BF% in 8–10 [112] and 6 [113] year-old children. On the contrary, increases in the fat mass index (FMI) were found following high-fat, low-fiber dietary patterns in 11–15-year-old children [114].

Other than weight, height is also a very important parameter to consider when measuring body composition in children, being, together with BF%, the main growth reference indicators [115]. Despite some studies reporting decreases in body fat percentage (BF%) after ingesting a quality diet, this was not observed in other parameters, such as BMI or FMI (fat mass (kg)/height2 (m)), in which height is of relevant importance [112] and could determine the development of obesity in later life stages [116]. Nevertheless, Voortman et al. (2016) [113] reported a non-independent association between dietary patterns and body composition due to similar results obtained for FMI and fat-free mass index (FFMI) with and without adjustment for height.

Although it has been shown that the use of body weight and height, as well as their relationship (BMI), presents important limitations, they are still used as reference values for body composition. In this sense, the most appropriate method would be to be able to differentiate adiposity from lean mass through its measurement from body composition [117]. However, the use of different anthropometric measurements, such as skin folds or girths, may contain errors because it does not necessarily reflect the total amount of fat in the body, due to a large percentage of body fat being internal [118]. Therefore, the use of dual-energy X-ray absorptiometry (DEXA) could provide reliable values of body composition [119] and be able to correctly relate them to different dietary patterns during childhood.

## 8. Dietary Patterns in Infancy and Oral Health

Oral health remains a considerable public health problem that needs greater effort, since it has not improved in the last decades [120]. There are several interactions between the nutrients we take and the oral health we have, and a balanced diet has been correlated to better quality and quantity of saliva, periodontal tissue, and dental elements [121]. The development of good habits to avoid oral health diseases is especially important during childhood [122]. Moreover, an imbalanced diet during this period of oral development can produce larger damage than in adult years [123].

Dental caries—the demineralization of the inorganic part of the tooth—is still the most prevalent oral health disease globally. It is caused by a bacterial activity enhanced by its anaerobic metabolism of sugars in the diet [124]. There is a scientific consensus on an excess intake of added sugars as the main reason to develop this disease [125]. Added sugars can be defined as those “found in foods other than grains, vegetables, whole fruit, and milk” [125], and a reduction of just 10% of sugar intake in children has been reported to give health benefits to12-year old children in Poland [126]. The use of fluorine has been found as an important way of caries reduction (approximately by 20–40%), although the prevalence of sugar intake keeps being a threatening behavior for oral health [127]. A higher sugar consumption (even with low levels) throughout life represents a higher risk of developing dental caries [128], and sugar consumption has been recommended to be lowered to just 2–3% of total energy intake [129].

An important question that arises from previous research is: what are the biggest determinants that cause children to have an excess intake of sugars? If we take into account the origin of added sugars in the diet, sugar-sweetened beverages lead the list, followed by desserts, cereals, and candies [130]. Not only the behaviors of children, but also the beliefs and conducts of their parents influence their diet [131]. The ease of purchase, the cost, the good taste, and its relationship to fruit flavors favored their prevalence [132]. The consumption of these products by their parents also enhanced their prevalence [133], and stores have been reported as the main place of purchase, rather than fast food restaurants or schools [134]. Peer intake [135] and media advertisements also have a great influence on children.

Another important behavioral pattern is the time in which the energy intake is happening. Snaking between the main meals fosters the continuous presence of sugars in the oral cavity, thus increasing the probability of oral disease [136]. There is also evidence of the relationship between eating before bedtime and a higher caries occurrence, probably due to the salivary flow decrease in sleep time [137].

Socioeconomic status also influences dietary patterns and their influence on oral health [138], and food insecurity has been linked to more overall sugar intake [139] and added sugar through drinks [140]. Not only the country of residence, but the background and origin of the families influence their dietary patterns and oral health [141]. Infections due to both malnutrition and poor oral health are still common in sub-Saharan Africa [142].

Despite the wide knowledge on the relationship between sugar intake and caries development, there is still a lack of knowledge on the best strategies to reduce added sugar intake in children [143]. The best approaches are to provide alternatives to sweetened beverages [144], and to develop long-term interventions not only in the school [145], but with the help of parents; however, banning sweetened beverages does not seem to work in the long-term, since it may raise the desire of the children to consume those products [146]. The increase of taxes for these products seems to reduce their consumption [147]. Screen time and sedentary behaviors seem to also be related to a higher occurrence of oral diseases, although this relationship should be better studied in the future [148].

## 9. Dietary Patterns in Infancy and Immune System

Nutrition is the most influential environmental factor during childhood growth. In this line, the pre-and post-natal dietary patterns are fundamental for the development of many physiological functions and immune responses [149]. The human immune system can generate two types of responses, innate and adaptive [150]: the innate immune system provides rapid defense (hours) against invading microbes, whereas the adaptive mounts a posterior defensive response (days) [151].

Moreover, the immune system has the ability to regulate the physiological response to food, as well as the metabolism of nutrition [152]. Some experimental results indicate that diet and its components are able to profoundly influence immune responses, thus affecting the development of inflammatory and autoimmune diseases [153]. Related to this, the development and function of immune cells rely on the type and amount of food consumed. Both over- and under-nutrition, including micronutrient deficiency, are linked to poor immune outcomes. Undernutrition is linked to greater morbidity and mortality from infections [154], whereas obesity is linked to increased infection rate, and is associated with immune disorders such as allergies and autoimmunity [155]. Specific metabolic pathways will be activated by the type of energy available to an immune cell, which can dictate whether it becomes pro- or anti-inflammatory. Thus, both the type and amount of food consumed have the potential to affect immunometabolism, and there is potential for diet to be used as a tool to fine-tune the immune response [153].

Additionally, the immune system is mediated by another determining element, the human microbiota, directly influencing health right from birth [156]. This parameter contains several bacteria, 10 times higher than the number of cells in the human body [157]. Therefore, several studies have reported that alterations in the gut microbiome in childhood are associated with an increased risk of developing diseases later in adulthood [158]. The intestinal microbiota intervenes in immune responses through the production of a series of metabolites that modulate the inflammatory signaling interacting both directly and indirectly with host immune cells [159]. Previous data have shown that the construction and maturation of the gut microbiome in children may be largely determined by nutrition in the early stages of life [160]. Therefore, the follow-up of healthy dietary patterns will determine the correct development of children through the direct influence it exerts on the intake of essential nutrients and its indirect effect on the composition of the intestinal microbiota [161].

In the prenatal stage, it has been shown that an unhealthy diet of the mother can lead to the development of defective lean gut microbiota in the fetus [162], resulting in a dysfunction of the immune system [163]. After birth, a key stage for the optimal development of the intestinal microbiota begins. However, this stage is characterized by presenting a very low variety of microbiota cells [164], which stabilizes in later stages (2–3 year-old), after the cessation of breastfeeding [165]. In this line, the breastfeeding period is considered one of the most important stages for the development of the newborn, due to breast milk being considered the best nutritional option for growth and healthy development, since it contains a wide range of microorganisms [166] and health-protective compounds [150]. To support this, a recent study reported that breastfed infants had a higher number of bacterial cells than formula-fed infants [167], decreasing the possibility of predisposing children to potential pathogens [157]. However, the introduction of solid food in the infant diet has been reported to be an important event influencing the microbiota as well [164], allowing a greater diversity and bacterial load [168].

However, other external factors, such as lifestyle, culture, and eating habits, can influence the proper functioning of the intestinal microbiota. According to dietary patterns, alterations in this parameter can cause changes in the intestinal microbial composition, increasing the risk of developing allergic diseases as a consequence of a decrease in the immune system [169]. However, adopting a varied diet with a diversity of allergens during the first years of life will reduce the risk of developing food allergies in the later stages of life [152]. In this line, Berding et al. [170] showed a significant correlation between the consumption of fiber-rich foods, such as vegetables, fruits, and grains, with a higher degree of microbiota stability. Thus, diet can be an effective strategy to promote health, by strengthening the immune system or preventing deviation of the microbiota in the early stages of life.

## 10. Types of Diet and Supplementation on Psychological Health in Infancy

Psychological health is still a matter of concern worldwide, but some indicators suggest that psychological diseases will continue to rise worldwide [171]. A healthy diet seems to have a potential role at the early stages of life to prevent mental health diseases over a lifetime [172], to prevent cognition disabilities [173], and to support intelligence [174].

Cognition abilities are the result of several complex brain functions, such as attention, memory, thinking, or learning, and they can be regulated by different factors, such as nutrition [175]. Children may be especially exposed to bad dietary habits, since the brain experiences a great evolution during infancy [176]. The importance of breakfast for children has been reported, since the glucose metabolism in the brain of children can be twice of adults’ [175], so it can help them to enhance their cognition [177]. Omega-3 fatty acids are often low in the Western diet, and scientific literature suggests positive associations between docosahexaenoic and eicosapentaenoic acids, and cognitive, visual, and motor functions in children [178]. Vitamin B12 and folic acid levels have been traditionally studied from prenatal stages and during infancy to help in brain development [179] and prevent depression pathogenesis [180]. A consensus on whether zinc supplementation favors cognitive function is still not clear, although scientific literature suggests that adequate zinc levels can improve neurocognition [181]. Vitamin B12 and vitamin D deficiencies and elevated values of homocysteine can be triggers of depression in children, although more research is needed, especially taking into account the earlier development of these kinds of diseases, from early adulthood to infancy [182].

The relationship between mental disorders and diet patterns has traditionally been assessed in adults [183]. The connections between unhealthy dietary patterns and worse mental health seem to be more consistent than the associations between healthy dietary patterns and better mental health [184]. A Western lifestyle and dietary pattern; stressful events and situations; and a preference for takeaway fast food, confectionery, and red meat had a positive relationship with depressive and aggressive behaviors, whereas higher consumption of vegetables and fresh fruit was significantly associated with better behavioral patterns in early adolescence [185]. A relationship was also found between increased consumption of high-sugar food and emotional instability in children [186], whereas higher consumption of fruit and vegetables was related to a better behavioral pattern in 4–12-year-old children in Australia [187]. The association between higher stress levels and an increase in unhealthy food has been highlighted as a matter of importance, and it is believed to happen quite early in life: in children of approximately 8–9 years old, this relationship starts [182]. Iron deficiency has been remarked as a common problem both in developed and developing countries, and children are especially vulnerable, since iron acts in key aspects in the nervous system development, such as brain metabolism, neurotransmitter homeostasis, and myelin production [188]. A lack of iron during infancy can lead to slight consequences [189], whereas a lack of iodine seems to influence mental development in 5-year-old children [190]. Western dietary patterns with a high energy consumption based on fat and sugars enhance inflammation and adiposity, which are related to a higher risk of suffering mental health problems, including depression, before adulthood [185]. Supplementation with minerals and vitamins has been also found to work well with ADHD children to regulate aggressive behaviors and emotion regulation [191], whereas a diet high in sugars and fats could enhance hyperactivity and disruptive behaviors in sharp contrast to higher consumption of fruits and vegetables [192].

Light but continuous malnutrition during the first 2 years of life negatively influences reasoning, intelligence, language, learning, and attention [193]. In early childhood, intelligence also seems to be related to a higher amount of consumption of home-prepared food, fruit, and vegetables during infancy. Higher consumption of fruits, salad, rice, and pasta, rather than processed food with high levels of fat and sugar, also positively influences intelligence in children [174].

## 11. Alternative Types of Diet in Infancy

Nutritional patterns and behaviors are continually altered and adapted due to a variety of factors that can be agglomerated into three factors: ideological factors, such as religious concerns [194], concerns regarding animal welfare [195], the use of antibiotics and hormones in livestock, and the excessive exploitation of environmental resources [196]; factors related to nutritional trends, such as ketogenic diets, paleo diets, gluten-free diets, diets based on “real food”, etc.; and finally, factors of economic origin. Chicken costs 8.8% more than it did in October 2020, and seafood was up by an average of 7.5% [197]. In contrast, fast food products and services maintain balanced prices in the markets [198]. This undoubtedly supposes the adoption and change in the behaviors of the consumer. Therefore, there is a growing number of parents who are adopting alternative dietary patterns and behaviors, which are far from the ideal nutritional pattern of the Mediterranean diet [199]. Interestingly, around 80% of the people who decide to adopt one of these nutritional patterns do so using the Internet as the main source of information, considering it as “reliable information” [200]. It is worrying when parents initiate this dietary pattern and drag their children into it without true knowledge of whether this diet could cause any harm to them.

Regarding a gluten-free diet, gluten is described as a mixture of storage proteins, prolamins, hordeins, and secalins, found in wheat, barley, and rye, respectively [201]. Celiac disease is a chronic systemic autoimmune disorder caused by an irreversible intolerance to gluten proteins in genetically susceptible individuals [202], with an estimated prevalence of 1%, and the only treatment is a lifelong strict gluten-free diet [203,204]. In this line, the consumption of gluten-free foods has significantly increased over the last 30 years [205], showing strong economic growth in this sector, and with a wide and varied spectrum of products. However, besides the documented positive effects of following a gluten-free diet for treating the spectrum of gluten-related disorders, there is no evidence of a gluten-free diet being a better option than a standard diet for the general population [206]. Furthermore, the belief of a gluten-free diet being a miracle diet for weight loss is without evidence [207]. Overweight or obese celiac disease patients who followed a gluten-free diet for 2 years [208] and 2.8 years [209] were observed to gain weight, whereas in a study conducted on children who followed a gluten-free diet for at least 12 months, the percentage of overweight children almost doubled [210]. In this line, studies show gluten-free products have a greater carbohydrate and lipid content [211,212], and less fiber [213]. Patients with celiac disease following a gluten-free diet have been shown to have a deficit of fiber, iron, calcium, vitamin D, magnesium, folate, niacin, vitamin B12, and riboflavin [214]. Thus, there is no evidence of the beneficial use of a gluten-free diet in a healthy population, even less so in children, and it should only be prescribed for the treatment of gluten-related disorders.

Finally, the ketogenic diet has also gained a lot of followers in recent years. It is a strict diet consisting of minimal carbohydrate and protein intake and increased fat intake, looking to mimic the body’s response to starvation by using fat as the primary energy source in the absence of an adequate dietary carbohydrate source looking to induce ketosis [215]. Etiologically, very low-carbohydrate diets may lack vitamins, minerals, fiber, and phytochemicals [216], and are often low in thiamin, folate, vitamin A, vitamin E, vitamin B6, calcium, magnesium, iron, and potassium [217], while also being poor in fiber, affecting the microbiota [218]. Furthermore, ketogenic diets may increase chronic disease risks due to higher ingestion of certain foods (red meat, processed meat, saturated fat) linked to cardiovascular disease, cancer, diabetes, and Alzheimer’s disease, whereas intake of protective foods (e.g., vegetables, fruits, legumes, whole grains) is typically decreased [219]. We can conclude that there is no evidence of the use of a ketogenic diet on healthy children, and it could compromise their future health, increasing the risk of developing chronic diseases.

Therefore, after the analysis of the most popular diets worldwide, we can conclude that there is no evidence of their use in healthy children, and their use should only be limited to concrete therapeutic cases such as celiac disease or intractable epilepsy, where there is far more evidence of their efficacy.

## 12. Specific Policies to Promote Healthy Dietary Patterns from Infancy

As demonstrated, the different influences that children receive during their childhood are multiple and varied. In trying to specify which policies may be needed to achieve the implementation of healthier habits, we believe it is important to distinguish those that are currently being implemented, the type of intervention, and where. Based on the type of intervention, it should be noted that governments may apply nutritional or lifestyle interventions. Generally, most strategies are focused on schools or on children’s environments, including parent’s behaviors, but neither of them are focused on both areas. The provision of healthy meals during school hours or the modification of the school food service were successful methods of achieving a higher consumption of fruits and vegetables in countries such as Denmark [220] or the United States [221,222]. In addition, it should be noted that when programs not only focus on improving a specific healthy habit, but rather on the development of nutritional strategies focused on the acquisition of a healthy lifestyle as a whole, which includes physical activity, they are more effective; both China and the Netherlands were able to carry out this type of program [223,224]. The duration of the intervention period varied depending on the study; however, the most successful programs had a duration equal to or greater than one academic year [225].

However, there are other locations where people suffering from lack of access and money constraints shape unhealthy habits, and policies may have been prosperous there too. Ghana has implemented a school feeding program that also improved children’s nutritional literacy and status, and even increased the school enrolment numbers [226].

Even though dietary recommendations were the same in essentials while differing in minor points in different countries, due to their difference in geographical environment and traditional culture [227], the problem persists. Although studies reveal which methods to implement, not only during school age but also taking into account the nutritional status of the parents, the root of the problem is more serious, and healthy diets are unaffordable to many people. To increase the affordability, the cost of nutritious foods must come down, and countries will need a rebalancing of agricultural policies and incentives towards more nutrition-sensitive investment and policy actions all along the food supply chain to reduce food losses and enhance efficiencies at all stages [228].

## 13. Conclusions

The present narrative review concludes that parents´ culture, as well as their socioeconomic status, influence a child´s health and development from preconception. Therefore, progenitors´ nutritional status and their access to proper and healthy foods and supplements will be crucial to children’s optimal growth even before pregnancy. Afterward, the inadequacy or imbalance of mothers’ diets, leading to underweight, overweight/obesity, and micronutrient deficiencies, might be the most relevant factors in this process. Dietary supplementation with iron, vitamin D, vitamin B12, or folic acid, and a balanced diet are accepted to prevent later deficiencies after birth. Regarding breastfeeding, it plays an essential role in reducing health problems and the risk of suffering from chronic diseases during the early stages of life, such as alterations in the immune system or cognitive disorders. In the early years, a healthy environment is necessary for optimal development and to be able to avoid typical Western diseases, such as obesity, depression, or dental caries. Continuous collaboration and effort are needed to convey the importance of eating a good diet at all stages of life. The studies presented in this paper shed light on the possible options for nutritional strategies that should be implemented to achieve this goal.

## Data Availability

All data are presented in the manuscript.

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
