# Peer review of "Infancy Dietary Patterns, Development, and Health: An Extensive Narrative Review"

_children, 2022, doi:10.3390/children9071072_

Round 1
Reviewer 1 Report
This review shows from multiple perspectives that childhood diet is deeply related to health status as an adult. In particular, it teaches, based on many examples, that the social, economic, and cultural factors of the mother are deeply related to childhood diet and nutritional status.
In addition, the relationship between food allergy and the immune system, the findings on trace elements, and the relationship between oral hygiene and psychoneurology are interesting and new findings.
If one wanted, it would be desirable to devote a chapter to discuss what specific policies are desirable to achieve better childhood diets. It should be mentioned that the social context, such as aging rates, fertility rates, and population composition, is quite different between the so-called developed countries of old, such as North America, Western Europe, and Japan, and the emerging countries. In addition, it would be even better to discuss policies in different regions such as Europe, Asia, North America, Latin America, and Africa.
Reviewer 2 Report
This extensive narrative review presented evidence on parents’ nutritional status and eating habits and associations between infancy dietary patterns and health.
Abstract:
The sentence stated in Lines 17-18 could be re worded. Understanding a child’s health as a state of complete physical, mental and social wellbeing… It is not clear what the authors try to convey after that sentence?).
Line 20, where it says “this process” which cprocess?
Overall this manuscript could be shortened. For example: Lines 72-75 says the same than lines 78-79
Line 169 replace with ZINC
Some of the subtitles do not align with the content. E.g. #5 Dietary patterns and child development, discuss nutrients and children obesity.
LINE 272, when it says 27 studies, are the authors citing a review article? (ref 78)
Subtitle #7 the first paragraphs appear to refer to subtitle #4 suggest to move it.
The main concern is the misuse of the concept of “dietary patterns”. Researchers do not present many articles actually assessing dietary patterns, yet in all the titles they use this term. This could be misleading. For example, in title #5 Dietary patterns in infancy and child development, in LINE 361 authors discuss a study on protein intake and in line 267 they mentioned micronutrients.
There is an overstatement in the conclusion section, LINE 643, there is not actually studies addressing nutritional strategies to achieve the goal. This is misleading. Authors presented evidence regarding what is associated with poorer outcomes, but there is not possible options on what can be done clearly describe through the article.
